# ABO blood group and COVID-19 severity: Associations with endothelial and adipocyte activation in critically ill patients

Sophie Stukas[1], George Goshua[2,3], Edward M. Conway[4], Agnes Y. Y. Lee[5], Ryan L. Hoiland[6,7], Mypinder S. Sekhon[8,9], Luke Y. C. Chen[10,11]*

1 Department of Pathology and Laboratory Medicine, Djavad Mowafaghian Centre for Brain Health, University of British Columbia, Vancouver, British Columbia, Canada, 2 Department of Internal Medicine, Section of Medical Oncology & Hematology, Yale School of Medicine and Yale Cancer Center, New Haven, Connecticut, United States of America, 3 Center for Outcomes Research and Evaluation, Yale New Haven Hospital, New Haven, Connecticut, United States of America, 4 Department of Medicine, Centre for Blood Research, University of British Columbia, Vancouver, British Columbia, Canada, 5 Division of Hematology, University of British Columbia, Vancouver, British Columbia, Canada, 6 Department of Cellular and Physiological Sciences, Faculty of Medicine, University of British Columbia, Vancouver, British Columbia, Canada, 7 Centre for Chronic Disease Prevention and Management, University of British Columbia, Kelowna, British Columbia, Canada, 8 Division of Critical Care, University of British Columbia, Vancouver, British Columbia, Canada, 9 Centre for Brain Health, University of British Columbia, Vancouver, British Columbia, Canada, 10 Division of Hematology, Dalhousie University, Halifax, Nova Scotia, Canada, 11 Division of Hematology, University of British Columbia, Vancouver, British Columbia, Canada

* lchen2@bccancer.bc.ca

## Abstract

### Background

ABO blood group has been implicated both in susceptibility to, and severity of, SARS-CoV-2 infection. The aim of this study was to explore a potential association between ABO blood group and severity of COVID-19 infection in critically ill patients and the following biological mechanisms: inflammatory cytokines, endothelial injury, and adipokines.

### Methods

We conducted a retrospective study of 128 critically ill COVID-19 patients admitted to Vancouver General Hospital from March 2020-March 2021. Outcomes including 28-day mortality, need for mechanical ventilation and length of intensive care unit (ICU) stay were compared between patients with A & AB blood type vs. B & O blood type. Likewise, serum inflammatory markers, markers of endothelial activation, and adipokines were compared.

### Results

The association between ABO and severity of disease was confirmed. Patients with A&AB blood group had more frequent ventilation requirements compared to patients with blood group B&O (N(%): 35 (71%) vs 41 (52%), p=0.041), higher total ICU mortality (14 (29%) vs 9 (11%), p=0.018), longer median ICU stay (days, median [interquartile range]: 10 [6-19], vs 7 [3-14], p=0.016) and longer median hospital stay (26 [14-36] vs. 17 [10-30] p=0.034). No association was found between ABO blood group and serum inflammatory

**Data availability statement:** Data cannot be shared publicly because of UBC CREB privacy requirements. Data are available from the UBC Institutional Data Access / Ethics Committee (contact via corresponding author, or Dr. Cheryl Wellington, cheryl.wellington@ubc.ca or the research office, dom.research@ubc.ca) for researchers who meet the criteria for access to confidential data.

**Funding:** LC is supported by a philanthropic gift from the Hsu & Taylor Family through the VGH & UBC Hospital Foundation. GG is supported by the NOMIS Foundation, Frederick A. DeLuca Foundation, Yale Cancer Center, Yale Bunker Endowment, and the National Institutes of Health (NIH), National Heart, Lung, and Blood Institute (NHLBI) grant 1K01 HL175220. The contents of this article are solely the responsibility of the authors and do not necessarily represent the official views of the funding sources.

**Competing interests:** The authors have declared that no competing interests exist.

cytokines or their receptors [IL-6, IL-1b, IL-10, TNF, sIL-6R, sgp130] measured within the first 10 days of ICU stay. No association was found between ABO and plasma markers of endothelial injury [Thrombomodulin, ADAMTS13, sP-Selectin, Factor IX, Protein C, Protein S, vWF]. Among the plasma adipokines, there were no differences between lipocalin-2, PAI-1 or resistin. Notably, however, median adipsin was higher in patients with A&AB blood group compared to O&B (16.3 [4.2-38.5] x10$^6$ pg/mL vs. 9.61 [3.0-20.8] x 10$^6$ pg/mL, p=0.048).

## Conclusions

This retrospective single-center study confirms an association between A and AB blood type with more severe COVID-19. While an underlying mechanism was not identified, the finding of higher adipsin levels in patients with type A/AB blood warrants further investigation in larger prospective studies.

## Introduction

ABO blood type consists of carbohydrate antigens at the extracellular surface of human red blood cells. Numerous studies examining the association between ABO blood group and both susceptibility to, and severity of, COVID-19 infection have been published. Specifically, several large case-control studies and meta-analyses have demonstrated that patients with blood group A are more susceptible to COVID-19 and that patients with blood group O are less susceptible [1–5]. Regarding severity of COVID-19 infection, the literature on ABO blood type is more heterogeneous. Some studies, including one from our own center in British Columbia, Canada, have shown that blood group A is associated with more severe disease, and blood group O with less severe disease [6–8], while others have not found an association [9,10]. These differences may result from heterogeneity of study design, population-based differences, or some combination thereof.

Aside from simple observational studies, the potential impact of ABO blood type on susceptibility to, and severity of, COVID-19 infection is supported by other lines of evidence. First, and perhaps most important, large genome-wide association studies (GWAS) show an association between ABO blood type and susceptibility to infection [11–14]. In these GWAS studies, the *ABO* (histo-blood group ABO system transferase) gene consistently emerges as the strongest signal within the susceptibility group of loci [15]. One GWAS study also identified a potential association between *ABO* and severity of disease but the authors were not able to exclude horizontal pleiotropic effects and concluded further study was needed [16].

Second, ABO blood type has been implicated in several other infections. The potential for coronaviruses to differentially affect subjects with different ABO blood types was suggested during an outbreak of SARS-CoV-1 in 2003, wherein group O patients were less susceptible [17]. In contrast, blood group O is a well established risk factor for Norovirus infection [18,19] while it is protective against hepatitis B [20]. Blood type AB is associated with more severe hemorrhagic fever in those exposed to Dengue virus [21]. Blood group O has also been found to be protective from severe *Plasmodium falciparum* infection, an association thought to be related to decreased parasite "resetting" of red cells [22].

Third, COVID-19 has highlighted the relationship between inflammation, thrombosis and coagulopathy [23]. ABO has been implicated in immunological/thrombotic processes such as heparin induced thrombocytopenia (HIT) wherein O blood group is a risk factor for HIT [OR 1.42, [24]]. Patients with type O blood have lower levels of coagulation factors such as Factor

VIII and von Willebrand factor. Non-O blood type is associated with increased risk of thrombosis in the general population and in cancer associated thrombosis [25,26].

To date, two hypotheses have been proposed to explain the association between decreased susceptibility to disease, and decreased disease severity in patients with type O compared to type A red cells: (1) the anti-A theory, and (2) association of ABO blood types with differential systemic inflammation or endothelial activation. Regarding the anti-A theory, patients with type O and type B blood have anti-A antibodies, which may have anti-viral properties [27]. Studies on SARS-CoV-1 suggested that anti-A may impair binding of the spike protein with ACE2 [28]. An anti-glycan immune response has been postulated to play a role in SARS-CoV-2 infection [29] One *in-vitro* study demonstrated that the SARS-CoV-2 receptor-binding domain preferentially recognizes blood group A, [30] but similarly convincing mechanistic studies *in vivo* are lacking.

Regarding the theory that differential systemic inflammation or endothelial activation due to ABO blood type, a wealth of studies have been published on the association between inflammation, hypercytokinemia (specifically IL-6) and outcomes in COVID-19 [31–35]. COVID-19 is also considered an endotheliopathy, with higher endothelial markers in more severe disease [36,37]. In general, von willebrand factor (vWF) is lower in patients with type O blood [38]. This is due to post-translational interactions between the blood group and VWF.

Further exploration of the mechanisms behind ABO blood group and disease severity may be helpful in several ways. At the height of the COVID-19 pandemic, risk stratification was based on simple clinical parameters (age, comorbidities, etc) as well as markers of inflammation such as CRP and IL6 [32,39–41]. ABO is a simple, widely available biomarker which may further help distinguish severe disease or those likely to develop severe disease. Confirming or refuting the association of ABO blood type with severity of disease, and determining the mechanism of such an association may provide insights applicable to future microbial pandemics [42]. Patients who are critically ill with respiratory failure have the worst outcomes and require the most resources. Identifying risk factors for intensive care admission remains a priority.

The objective of our study was two-fold. First, we conducted a retrospective study to re-examine the association between ABO blood type and severity of disease in a larger cohort of critically-ill patients with COVID-19. Specifically, we compared key clinical interventions required, including the need for mechanical ventilation, continuous renal replacement therapy (CRRT), and extracorporeal membrane oxygenation (ECMO), and ICU-related outcomes including length of stay and mortality. Second, we aimed to elucidate potential mechanistic differences between blood groups by comparing the serum/plasma levels of key inflammatory cytokines, biomarkers of endothelial injury, and adipokines quantified within the first 10-days of ICU stay.

## Methods

The study was approved by the University of British Columbia Clinical Research Ethics Board (UBC CREB, H20-00971) and registered on ClinicalTrials.gov (NCT04363008). Due to the retrospective nature of the present study, the University of British Columbia Clinical Research Ethics Board waived the need for obtaining individual informed consent. The study data were accessed Mar 30, 2020 to Sep 30, 2024. All research was conducted in accordance with the principles of the Helsinki declaration and Strengthening the Reporting of Observational Studies in Epidemiology guidelines.

## Study design and participants

Patients in the present study were enrolled as part of a prospective COVID-19 biomarker study [43,44]. Adult patients admitted to the intensive care unity (ICU) at the Vancouver

General Hospital with a diagnosis of pneumonia secondary to SARS-CoV-2 infection between March 30, 2020 and March 31, 2021 were included. Referral and admission to the ICU was made at the discretion of the attending intensivist in accordance with the Surviving Sepsis Campaign and provincial management guidelines at the time [45], including all patients requiring mechanical ventilation and once the non-invasively administered oxygen requirements exceed 6 liters/min with a peripheral oximetry saturation of < 94% as described [6]. Patients were excluded if: COVID-19 was an incidental finding on or during admission (e.g., admitted to ICU primarily for trauma), if COVID-19 was determined to be nosocomial in origin, or if study enrollment took place > 10 days after initial ICU admission. Clinical management was in accordance with the provincial treatment guidelines for critically ill patients with COVID-19 set forth by the British Columbia COVID Therapeutics Committee and included corticosteroids after June 2020 [46], and tocilizumab after Jan 2021 [47,48]. The JAK inhibitor baricitinib was introduced in British Columbia May 2021, after the end of enrolment for the present study [49].

## Outcomes and procedures

Demographics pertaining to: age, sex, body mass index (BMI), medical comorbidities, smoking (current status), symptoms upon presentation to the ICU and date of: symptom onset, hospital and ICU admission and discharge or death, research study enrollment, initiation and cessation of mechanical ventilation and veno-venous extracorporeal membrane oxygenation (VV-ECMO) were recorded. The following interventions were recorded if they took place during the participants' ICU stay: use of steroids or other immunomodulatory agents, including tocilizumab, mechanical ventilation, VV-ECMO, acute respiratory distress syndrome (ARDS) as defined by the Berlin criteria [50], and continuous renal replacement therapy (CRRT). Outcomes recorded included both 28-day and all ICU mortality, as well as duration of mechanical ventilation, ICU and hospital stay. Clinical laboratory values included: complete blood count (WBC count and differential, hemoglobin concentration, and platelet count), partial thromboplastin time (PTT), d-dimer, INR, liver enzymes, bilirubin, creatinine, ferritin, and C-reactive protein (CRP).

As a part of the research study, arterial blood samples for both EDTA plasma and serum isolation were obtained upon ICU admission and study enrollment. Analyses of all endothelial and adipocyte markers were carried out in EDTA plasma by Eve Technologies Corporation (Calgary, Alberta, Canada) in May of 2021. Factor IX, protein C, protein S and Von Willebrand factor (vWF) were quantified using Custom Human Coagulation Panel 3 4-Plex Assay (cat ID HPROCOAG3-04-21); ADAMTS13 and sP-Selectin were quantified using Custom Human Cardiovascular Disease Panel 2 2-Plex Discovery Assay Array (cat ID HDSAA6); thrombomodulin was quantified using Human Cardiovascular Disease Panel 4 1-Plex Assay (no cat ID available); and adiponectin, adipsin, lipocalin-2, plasminogen activator inhibitor-1 (PAI-1), and resistin were quantified using the Custom Human Adipokine Panel 1 5-Plex Assay (no cat ID available). Serum IL-6, IL-10, and TNF were quantified using the Simoa Cytokine 3-Plex A Advantage Assay (cat ID 101160) and IL-1β was quantified using the Simoa IL-1β Advantage Assay (cat ID 101605) on the HD-1 platform from Quanterix Inc. (Billerica, MA, USA) as previously described [35,44], from Apr 2020 to Mar 2021. Serum soluble IL-6 receptor (sIL-6R) and soluble gp130 were quantified using commercially available ELISAs from R&D Systems Inc. (Minneapolis, MN, USA; cat ID DR600 and DGP00) between June-Jul 2021. All cytokine analyses were done in the lab of Dr. Cheryl Wellington at the University of British Columbia. For the inflammatory markers only (IL-6, IL-10, TNF, IL-1β, sIL-6R and gp130), participants already on Tocilizumab at the time of the research blood

draw were excluded from analysis given its known effect on these markers. However, nearly all patients enrolled after June 2020 and included in this study received dexamethasone prior to ICU admission. Each assay plate contained an 8-point calibrator curve and 2-3 control samples to monitor inter-plate variability (inflammatory analysis only). All samples were analyzed in duplicate, with the average of the values reported, in a randomized order. The average intra-sample CV was: Human Coagulation Panel 3 4-Plex Assay 8-20%; Human Cardiovascular Disease Panel 2 2-Plex 14-19%; Human Cardiovascular Disease Panel 4 1-Plex 9%; Human Adipokine Panel 1 5-Plex Assay 9-26%; Simoa Cytokine 3-Plex Assay 5-6%; Simoa IL-1β assay 14%; and sIL-6R and sgp130 8%.

## Statistical analysis

Data normality was assessed using a Sharpiro-Wilk test and Q-Q plots. As biomarker data was almost exclusively non-normally distributed, continuous data were described using median and interquartile range (IQR) and group differences between participants with blood group A or AB were compared to participants with blood group B or O using a non-parametric Mann-Whitney U test. Categorical variables were described using N and frequency and group differences were tested using a Fisher's exact test. For categorical clinical outcomes, such as the need for mechanical ventilation, CRRT, or ECMO and mortality, odds ratios (OR) were determined using a logistic regression model corrected for age and sex.

## Results

Between March 30, 2020 and March 31, 2021, a total of 132 patients were admitted to the ICU due to COVID-19 and enrolled into the research study, 128 of whom met the additional criteria of enrollment and research blood collection within 10 days of initial ICU admission. The four participants excluded from the study were transferred from other hospitals for the purposes of VV-ECMO, and thus ranged between 22-82 days from initial ICU admission and study enrollment. Of the 128 participants, 25 were included in our previous report examining the relationships between ABO and severity of COVID-19 [6]. ABO grouping analysis showed that 45 participants were blood group A (35%), 4 were blood group AB (3.1%), 30 were blood group B (23%), and 49 were blood group O (38%). Participants were then stratified based on the absence or presence of anti-A antibodies into blood group A and AB (n = 49, 38%) versus blood group O and B (n = 79, 62%) for further analysis.

Overall, there were no statistically significant differences in the cohort demographics, co-morbidities, symptomatic presentation to the ICU or study enrollment between participants with blood group A or AB versus B or O (Table 1). On average, participants were predominantly male (~60%) in their mid-late 60s, with 86% presenting with at least 1 comorbidity, the most frequent of which was hypertension. Participants were admitted to the ICU an average of 7 days following the onset of symptoms, and were typically enrolled in the research study within the following 48 hours. Likewise, there were not any significant differences in the clinical laboratory measures taken upon ICU admission and study enrollment based on ABO grouping (S1 Table). The only exception to this was lower levels of d-dimer in participants with blood group A or AB compared to blood group B or O (1031 vs 1497 ug/L; p = 0.03). As expected, all participants with COVID-19 had lymphopenia, with elevated d-dimer, AST, ALT, LDH, CRP, and ferritin compared to reference ranges, consistent with previous reports of critically ill patients with COVID-19 [51,52] Clinical interventions and outcomes are shown in Table 2. Participants with blood group A and AB had a significantly higher rate of ventilation (71% vs 52%, p = 0.041; OR 2.42 95% CI 1.13 to 5.35), total ICU mortality (29% vs 11%, p = 0.018, OR 3.01 95% CI 1.12 to 8.42), length of ICU stay (median 10 vs 7 days, p = 0.016),

**Table 1. Comparison of demographics and clinical presentation for COVID-19 patients admitted to VGH ICU between March 30 2020 and March 31 2021 based on ABO blood group.**

| | A&AB (n = 49) | B&O (n = 79) | P-values[a] |
|---|---|---|---|
| **Demographics** | | | |
| Male, n (%) | 28 (56) | 49 (61) | 0.71 |
| Female, n (%) | 21 (43) | 30 (38) | |
| Age, y, median [IQR] | 68 [56, 77] | 62 [51, 74] | 0.25 |
| BMI, kg/m², median [IQR] | 32 [26, 34] | 29 [27, 33] | 0.77 |
| *Comorbidities, n (%)* | | | |
| HTN | 24 (49) | 40 (51) | >0.99 |
| Diabetes | 20 (41) | 31 (39) | >0.99 |
| Obesity | 7 (14) | 5 (6) | 0.21 |
| Dyslipidemia | 14 (29) | 35 (44) | 0.093 |
| CKD | 4 (8) | 8 (10) | >0.99 |
| CAD | 4 (8) | 13 (17) | 0.19 |
| COPD | 5 (10) | 9 (11) | >0.99 |
| Smoking | 12 (26) | 13 (17) | 0.25 |
| At least one co-morbidity | 7 (86) | 11 (86) | >0.99 |
| **Presentation to ICU** | | | |
| *Presenting symptoms, n (%)* | | | |
| Fever | 41 (84) | 59 (0) | 0.37 |
| Cough | 45 (92) | 70 (90) | 0.77 |
| Dyspnea | 35 (92) | 62 (95) | 0.67 |
| Myalgias | 6 (16) | 3 (5) | 0.07 |
| Diarrhea | 15 (39) | 32 (49) | 0.41 |
| Headache | 6 (12) | 11 (14) | >0.99 |
| *Enrollment information* | | | |
| Duration, days, med [IQR], symptoms and ICU admission | 7 [5, 9] | 7 [6, 10] | 0.17 |
| Duration, days, med [IQR], ICU admission and study enrollment | 1 [1, 2] | 1 [1, 2] | 0.22 |

ARDS, acute respiratory distress syndrome; BMI, body mass index; CAD, coronal artery disease; CKD, chronic kidney disease; COPD, chronic obstructive pulmonary disease; CRRT, continuous renal replacement therapy; HTN, hypertension; VV-ECMO, veno-venous extracorporeal membrane oxygenation.

[a]Pair-wise comparisons were conducted using a Mann Whitney U Test (continuous variables) or Fisher's Exact test (categorical variables)

and length of hospital stay (median 26 vs 17 days, p = 0.034), compared to participants with blood group B or O, consistent with our previous findings [6].

Next, we compared the circulating levels of inflammatory, endothelial, and adipokine markers based on ABO group taken upon ICU admission and study enrollment (Figs 1–3 and S2 Table). We started by looking at a core set of inflammatory cytokines, IL-1β, IL-10, TNF, and IL-6, as well as two of the receptor components that are critically involved in IL-6 signalling, sIL-6R and sgp130 (Fig 1). Consistent with our previous report [6], there were no differences in the inflammatory markers or signalling receptors between patients with A/AB and O/B blood groups. Next, we assessed a broad panel of endothelial markers, including soluble thrombomodulin, vWF and ADAMTS13, the enzyme response for vWF cleavage, sP-selectin, factor IX, protein C, and protein S (Fig 2). Again, there were no significant differences in any of the endothelial markers based on ABO grouping. Last, we tested a panel of adipokines, including adiponectin, adipsin, resistin, lipocalin-2, and PAI-1 (Fig 3). Notably, the median

**Table 2. Comparison of ICU interventions, complications and outcomes for COVID-19 patients admitted to VGH ICU between March 30 2020 and March 31 2021 based on ABO blood group.**

| | A&AB (n = 49) | B&O (n = 79) | P-values[a] | Odds Ratio[b] | 95% CI |
|---|---|---|---|---|---|
| **ICU interventions and outcomes** | | | | | |
| *ICU interventions and complications, n (%)* | | | | | |
| Steroids* | 45 (92) | 71 (90) | 0.49 | 1.38 | 0.40, 5.52 |
| Steroids* + Tocilizumab | 13 (27) | 29 (37) | 0.25 | 0.62 | 0.28, 1.35 |
| Ventilated | 35 (70) | 41 (52) | **0.041** | 2.42 | 1.13, 5.35 |
| VV-ECMO | 4 (8) | 4 (5) | 0.48 | 1.93 | 0.43, 8.82 |
| CRRT | 9 (18) | 7 (9) | 0.18 | 2.21 | 0.75, 6.79 |
| ARDS | 25 (51) | 32 (41) | 0.28 | 1.72 | 0.82, 3.66 |
| *Outcomes* | | | | | |
| 28-day ICU mortality, n (%) | 9 (18) | 7 (9) | 0.17 | 1.91 | 0.60, 6.29 |
| Total ICU mortality, n (%) | 14 (29) | 9 (11) | **0.018** | 3.01 | 1.12, 8.42 |
| Length of ventilation, days, median [IQR] | 10 [5, 15] | 8 [4, 18] | 0.91 | | |
| Length of ICU stay, days, median [IQR] | 10 [6, 19] | 7 [3, 14] | **0.016** | | |
| Hospital stay, days, median [IQR] | 26 [14, 36] | 17 [10, 30] | **0.034** | | |

*As of June 2020 dexamethasone became part of standard of care

[a]Pair-wise comparisons were conducted using a Mann Whitney U Test (continuous variables) or Fisher's Exact test (categorical variables)

[b]Odd's Ratios and 95% CI were generated using a logistic regression model corrected for age and sex where OR > 1 indicates an increased probability of an event occurring during the study period in blood groups A/AB vs O/B.

VV-ECMO, veno-venous extracorporeal membrane oxygenation; ARDS, acute respiratory distress syndrome; CRRT, continuous renal replacement therapy

levels of the complement system-triggering serine protease, adipsin, were 1.7-fold higher in patients with A&AB blood group compared to O&B: 16.3 x$10^6$ pg/mL [IQR 4.2-38.5 x$10^6$] vs. 9.61 x $10^6$ pg/mL [IQR 3.0- 20.8 x$10^6$ pg/mL], p = 0.048. There was also a trend toward higher median adiponectin: 4.82x$10^8$ pg/mL [1.96, 7.92 x$10^8$] vs 3.13 x $10^8$ [1.07, 5.95 x$10^8$], p = 0.066.

## Discussion

In this study, we investigated 1) whether differences in markers of endothelial and adipocyte activation are present in patient with A/AB blood group compared to O/B, and 2) extended on our group's previously reported association between ABO blood group and severity of COVID-19 in a larger cohort [6]. Our findings confirm that ABO blood group is associated with more severe disease, with a higher odds ratio of mechanical ventilation in A&AB vs B&O patients (2.42, p = 0.041) and total ICU mortality (3.01, p = 0.018) and both median length of ICU stay (10 days vs 7 days, p = 0.016) and hospital stay (26 days vs 17 days, p = 0.034). In contrast to our earlier study, we did not find a significant difference in need for renal replacement therapy (CRRT; OR 2.21, p = 0.18). We investigated underlying mechanisms to explain the association between severity of COVID-19 infection and ABO blood group by comparing three panels of blood biomarkers between patients with A/AB vs. B/O blood group: 1) serum cytokines and IL-6 signaling receptors (Fig 1); 2) plasma markers of endothelial function and activation (Fig 2); and 3) plasma adipokines (Fig 3).

### 1) Serum cytokines and IL-6 signaling receptors

Although elevated IL-6 is considered a key prognostic factor in severe COVID-19 [31], and inhibition of IL-6 reduces mortality [47,53] in the present study, no differences were found in the circulating levels of this cytokine or IL-6 signaling receptors (IL-1β, IL-10, TNF, IL-6,

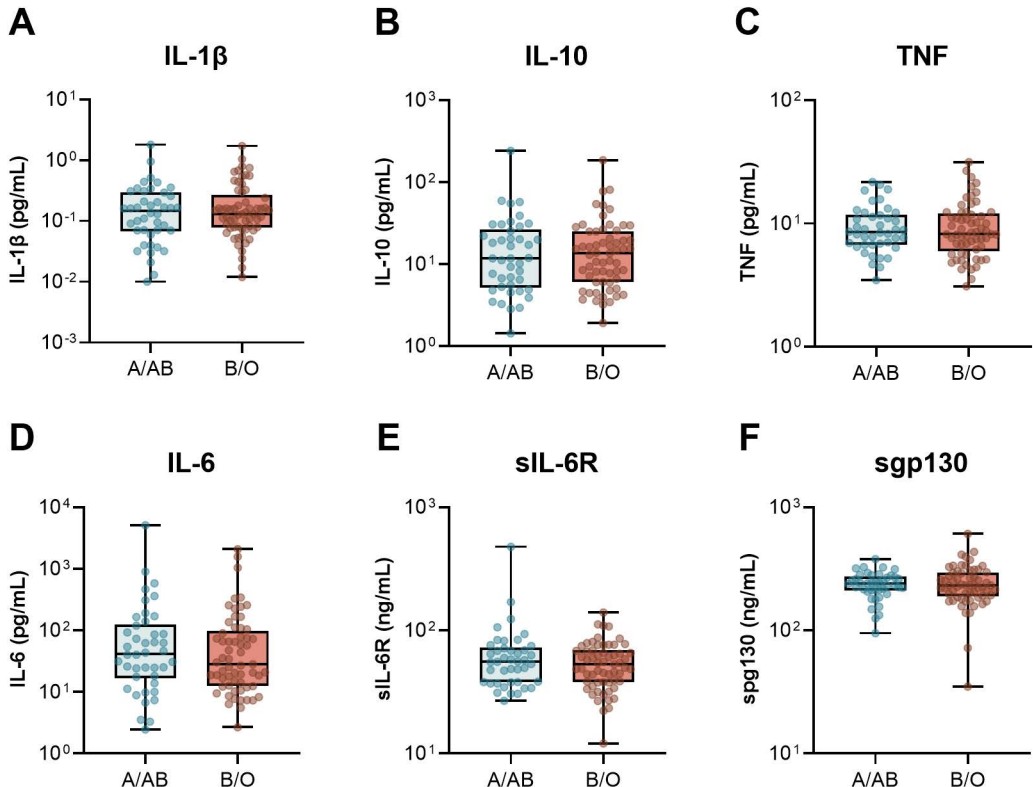

**Fig 1. Serum cytokines and IL-6 signaling receptors in critically ill patients with COVID-19 based on ABO grouping.** The levels of A) IL-1β, B) IL-10, C) TNF, D) IL-6, **E)** sIL-6R, and **F)** sgp130 were quantified in serum samples taken following ICU admission and study enrollment in 42 participants with blood group A/AB and 61 participants with blood group B/O. Box and whisker plot displays median, IQR, and min to max. Raw data are displayed on a log scale for better visualization. For tabular representation see S2 Table.

sIL-6R, sgp130). C-reactive protein (85 mg/L A&AB vs 78 mg O&B) and IL-6 levels (42.0 pg/L vs 28.3 pg/L) in this study were in keeping with much of the extant literature. Ours is one of the few studies to examine the other major components of the IL-6 trans signaling pathway [54,55]. We found no ABO blood type-dependent differences in sIL-6R (55.8 ng/L vs 53.2 ng/mL) or sgp130 (240 ng/L vs 232 ng/L). Likewise, a Swedish study found elevated IL-6 levels in patients with severe vs moderate COVID-19 but no differences in sIL-6R or sgp130 [56]. In contrast, a study of 104 Italian patients compared to healthy controls from the first wave of the COVID-19 pandemic demonstrated elevated levels of IL-6 (265.5 pg/mL vs 1.92 pg/mL, p < 0.0001) and sIL-6R (39.71 ng/mL vs 30.01 ng/mL p < 0.005), and lower levels of the buffer sgp130 (181.5 ng/mL vs 324.3 ng/mL, p < 0.0001) in those with COVID-19 [57]. Rodrĭguez-Hernẵndez et al. reported a Spanish study of 366 patients hospitalized with COVID-19 found that elevated IL-6 ≥ 27.4 pg/mL (HR 2.27), decreased sIL-6R ≤ 34.5 ng/mL (HR 2.50), and decreased sgp130 ≤ 367.5 ng/mL (HR 2.25) were risk factors for mortality [58].

## 2) Plasma markers of endothelial function and activation

Previous studies have demonstrated a strong association between markers of endothelial inflammation and COVID-19 severity in patients hospitalized for incident COVID-19 in the intensive care unit (ICU) versus non-ICU floors versus controls [36,37]. Moreover, pulmonary vasculopathy was a significant finding in early autopsy studies demonstrating that blood

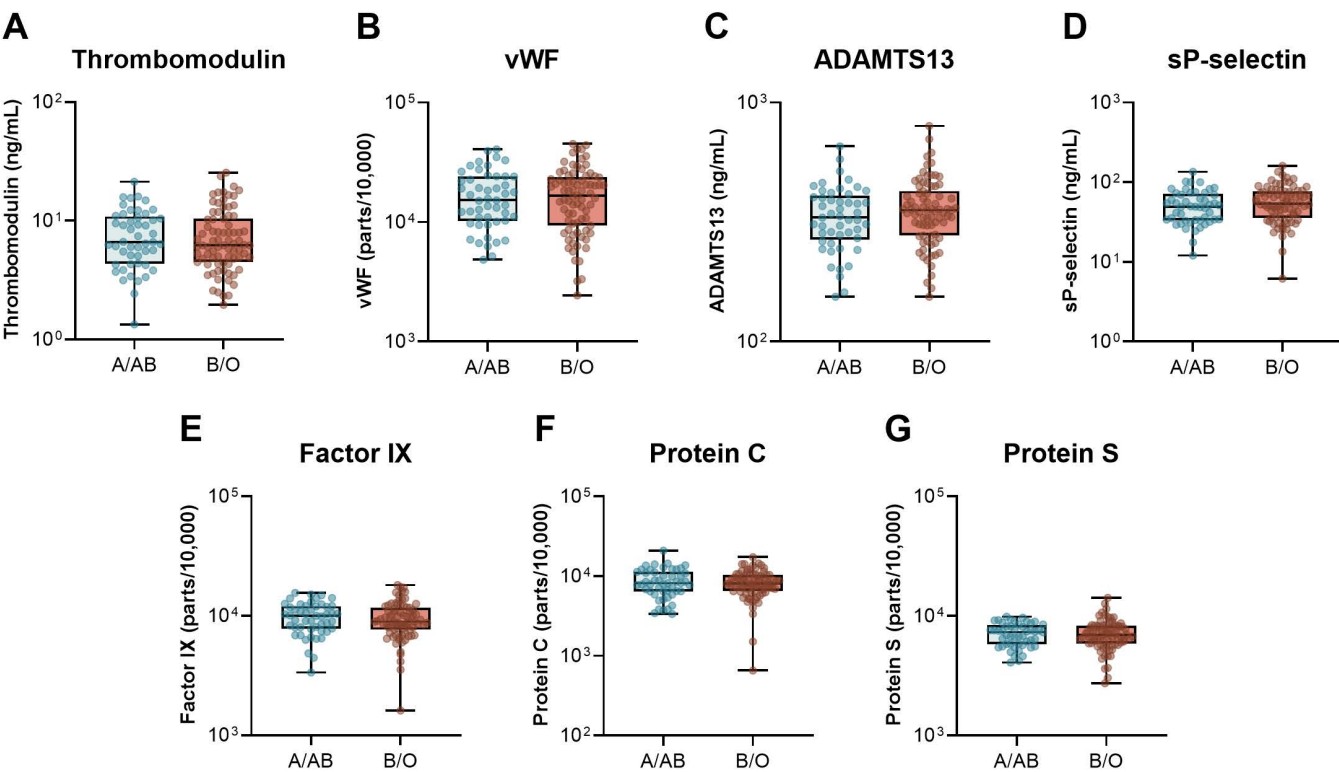

**Fig 2. Plasma markers of endothelial function and activation in critically ill patients with COVID-19 based on ABO grouping.** The levels of **A)** thrombomodulin, **B)** vWF, C) ADAMTS13, **D)** sP-selectin, **E)** factor IX, **F)** protein C, and **G)** protein S were quantified in plasma samples taken following ICU admission and study enrollment in 49 participants with blood group A/AB and 79 participants with blood group B/O. Box and whisker plot displays median, IQR, and min to max. Raw data are displayed on a log scale for better visualization. For tabular representation see S2 Table.

vessels are substantially impacted in severe COVID-19 [59]. However, these studies did not examine the effect of blood type on endotheliopathy. While patients with type O blood have lower von Willebrand factor at baseline, in the critical care context we did not find an association between blood type and markers of endothelial activation (thrombomodulin, vWF, ADAMTS13, sP-selectin, FIX, protein C, Protein S) in this study. D-dimer in participants with blood group A or AB was lower than those with blood group B or O (1031 vs 1497 ug/L; p = 0.03); whether this finding has clinical significance is unclear. The differential effect of blood type on outcomes in severe COVID-19 may be independent of the prior reported effect of endotheliopathy on these same outcomes (i.e., severity of COVID-19 and mortality secondary to it). Taken together, these findings tend to dispel the ABO-endotheliopathy theory as contributory to adverse outcomes in COVID-19.

## 3) Serum adipokines

Obesity is associated with increased mortality in COVID-19 [60,61]. Likewise, elevated adipokines are associated with more severe disease [62]. In the present study, no differences in resistin, lipoalin-2, and PAI-1 were found between A/AB and B/O patients. A systematic review of eight studies examining adipokines in COVID-19 concluded that adiponectin levels were significantly increased in patients with severe compared to mild COVID-19 [63]. In the present study, adiponectin was modestly elevated in A/AB compared to B/O, although not to a statistically significant extent ($4.82 \times 10^8$ vs $3.13 \times 10^8$, p = 0.0661).

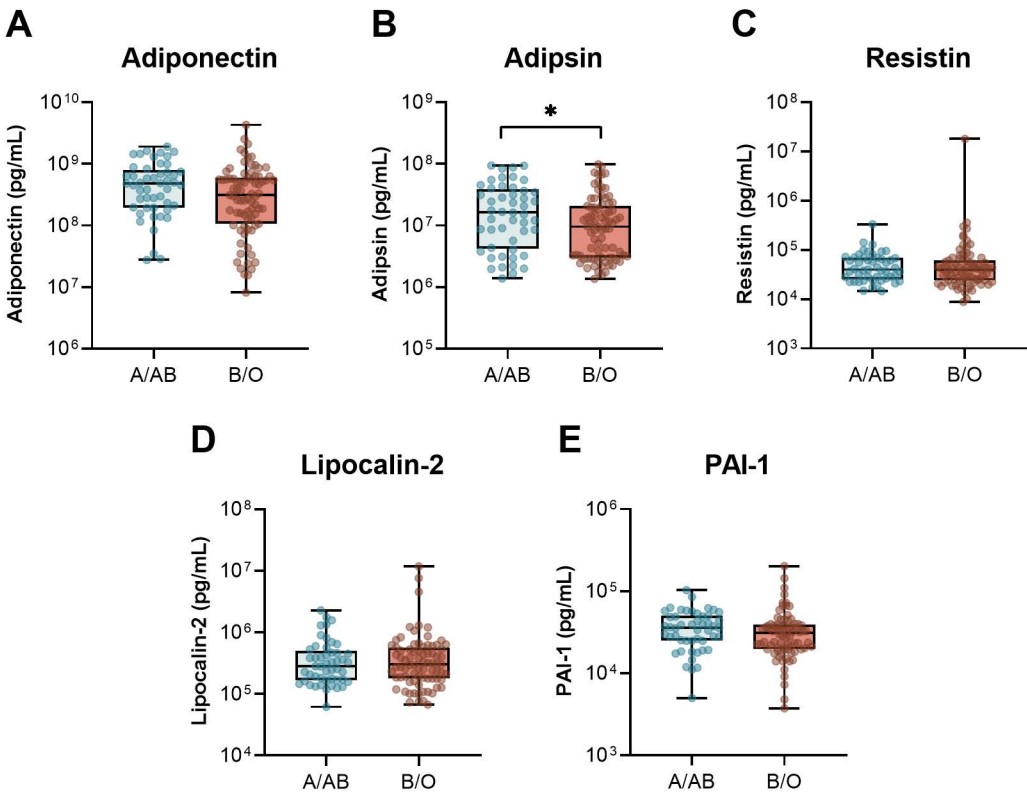

**Fig 3. Plasma adipokines in critically ill patients with COVID-19 based on ABO grouping.** The levels of **A)** adiponectin, **B)** adipsin, **C)** resistin, **D)** lipocalin-2, and E) PAI-1 were quantified in plasma samples taken following ICU admission and study enrollment in 49 participants with blood group A/AB and 79 participants with blood group B/O. Box and whisker plot displays median, IQR, and min to max. Raw data are displayed on a log scale for better visualization. * Represents $P < 0.05$. For tabular representation see S2 Table.

Median adipsin levels were higher in A&AB vs B&O ($1.63 \times 10^7$ pg/mL vs $9.61 \times 10^6$, $p = 0.048$). Adipsin is primarily synthesized by adipocytes. It is a serine protease, also known complement factor D. It is highly specific, in that it cleaves complement factor B, thereby activating complement, a major arm of innate immunity [64–66]. Interestingly, adipsin/factor D may also improve β Cell function in diabetes [67]. Adipsin and leptin are elevated in children with COVID-19 compared to convalescent samples and healthy controls [68] An earlier study from our center with an overlapping cohort of 25 critically ill patients also in the present, study demonstrated that elevated adipsin/factor D levels were associated with higher mortality [69]. These findings are thus intriguing, and raise the question of a role for complement and this particular complement-triggering serine protease, in the ABO-dependent differences in COVID-19 outcomes [70,71].

## Limitations and future work

This was an exploratory, observational study. In spite of an enrollment of > 100 patients, it is possible that the sample size was insufficient to detect small differences in the blood biomarkers examined. Inflammatory markers in severe COVID-19 are very dynamic, and we attempted to capture patients in the same phase of disease course by targeting sample collection to within the first 10 days of ICU stay. Nonetheless, markers might peak earlier in the course of the disease and are undoubtedly modified by interventions such as corticosteroids.

In spite of these limitations, exploring the potential interactions between novel viruses and easily identifiable biological risk factors such as the ABO blood groups will be important for future viral pandemics in helping to risk stratify patients and possibly lead towards tailored or targeted therapeutics. Indeed, a recent study has demonstrated that the discrepancy between GWAS studies and clinical studies of ABO and severity of disease may be related to differences in mRNA and protein expression of genes [72]. Investigating ABO blood group associations with COVID-19 in larger, multi-center studies may yield other biomarkers of interest. Last, we acknowledge the potential influence of SARS-CoV-2 variant on our results. While we do not have genetic data on our whole cohort, it is highly likely that the majority of our participants (>80%) were infected with the wild-type strain of SARS-CoV-2 given their admission date to the ICU (prior to April 2021) and the timing of the third wave and appearance of the alpha variant of concern in the late spring of 2021 in British Columbia.

## Conclusion

This study confirms the association between A & AB blood group and severity of COVID-19 compared to O & B. No differences were found in levels of inflammatory cytokines, markers or endothelial activation to account for the ABO-dependent findings. There was a statistically higher level of the adipokine adipsin in patients with A & AB blood groups and examination of adipokines in larger cohorts is warranted.

## Supporting information

**S1 Table. Clinical laboratory measures [taken upon study enrollment] for COVID-19 patients admitted to VGH ICU between March 30 2020 and March 31 2021 based on ABO blood group.**
(DOCX)

**S2 Table. Research laboratory measures [taken upon study enrollment] for COVID-19 patients admitted to VGH ICU between March 30 2020 and March 31 2021 based on ABO blood group.** For graphical representation see Figs 1–3.
(DOCX)

## Acknowledgments

The authors thank Dr. Cheryl Wellington for her expert advice and assistance with cytokine measurement.

## Author contributions

**Conceptualization:** Sophie Stukas, George Goshua, Edward M. Conway, Agnes Y.Y. Lee, Ryan L. Hoiland, Mypinder S. Sekhon, Luke Y.C. Chen.

**Data curation:** Sophie Stukas, Luke Y.C. Chen.

**Formal analysis:** Sophie Stukas, George Goshua, Luke Y.C. Chen.

**Funding acquisition:** Luke Y.C. Chen.

**Investigation:** Sophie Stukas, George Goshua, Luke Y.C. Chen.

**Methodology:** Sophie Stukas, George Goshua, Luke Y.C. Chen.

**Project administration:** Luke Y.C. Chen.

**Resources:** Edward M. Conway, Mypinder S. Sekhon, Luke Y.C. Chen.

**Supervision:** Edward M. Conway, Luke Y.C. Chen.

**Validation:** Luke Y.C. Chen.

**Writing – original draft:** Sophie Stukas, George Goshua, Edward M. Conway, Agnes Y.Y. Lee, Ryan L. Hoiland, Mypinder S. Sekhon, Luke Y.C. Chen.

**Writing – review & editing:** Sophie Stukas, George Goshua, Edward M. Conway, Agnes Y.Y. Lee, Ryan L. Hoiland, Mypinder S. Sekhon, Luke Y.C. Chen.

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
