## [Decision Letter · Decision Letter 0]

13 Dec 2024

PONE-D-24-45720Association of ABO Blood Group with Endothelial and Adipocyte Activation in COVID-19PLOS ONE

Dear Dr. Y.C. Chen,

Thank you for submitting your manuscript to PLOS ONE. After careful consideration, we feel that it has merit but does not fully meet PLOS ONE’s publication criteria as it currently stands. Therefore, we invite you to submit a revised version of the manuscript that addresses the points raised during the review process.

We look forward to receiving your revised manuscript.

Kind regards,

Santosh K. Patnaik, MD, PhD

Academic Editor

PLOS ONE

Journal Requirements:

2. Thank you for stating the following financial disclosure: [This work was supported by a philanthropic donation from the Hsu & Taylor Family through the VGH & UBC Hospital Foundation.]. Please state what role the funders took in the study. If the funders had no role, please state: "The funders had no role in study design, data collection and analysis, decision to publish, or preparation of the manuscript." If this statement is not correct you must amend it as needed. Please include this amended Role of Funder statement in your cover letter; we will change the online submission form on your behalf.

Additional Editor Comments:

The manuscript has been examined by five referees. I request that you kindly address all of their concerns through changes in the manuscript and/or comments in the Response to Review letter.

Reviewers' comments:

Reviewer's Responses to Questions

**Comments to the Author**

1. Is the manuscript technically sound, and do the data support the conclusions?

Reviewer #1: Partly

Reviewer #2: Partly

Reviewer #3: Yes

Reviewer #4: Yes

Reviewer #5: Yes

2. Has the statistical analysis been performed appropriately and rigorously? 

Reviewer #1: Yes

Reviewer #2: Yes

Reviewer #3: Yes

Reviewer #4: Yes

Reviewer #5: Yes

3. Have the authors made all data underlying the findings in their manuscript fully available?

Reviewer #1: Yes

Reviewer #2: Yes

Reviewer #3: Yes

Reviewer #4: Yes

Reviewer #5: Yes

4. Is the manuscript presented in an intelligible fashion and written in standard English?

Reviewer #1: Yes

Reviewer #2: No

Reviewer #3: Yes

Reviewer #4: Yes

Reviewer #5: Yes

5. Review Comments to the Author

Reviewer #1: This manuscript explores the association between ABO blood groups and COVID-19 severity, focusing on clinical outcomes in critically ill patients and the role of endothelial and adipocyte activation markers. The study addresses an important question about potential risk factors for severe COVID-19 outcomes, and the findings on blood group A/AB being associated with higher ICU mortality and ventilation requirements are intriguing and could have implications for clinical risk stratification. However, certain areas require clarification and enhancement to improve the manuscript’s clarity, scientific rigor, and relevance to a global audience.

Title and Abstract

Title Suggestions: The current title, “Association of ABO Blood Group with Endothelial and Adipocyte Activation in COVID-19,” could be made more descriptive. Consider revising the title to capture both the critical patient population (critically ill COVID-19 patients) and the study’s specific objectives (COVID-19 severity, endothelial/adipocyte activation). A suggested title might be, “ABO Blood Group and COVID-19 Severity: Associations with Endothelial and Adipocyte Activation in Critically Ill Patients.”

Abstract Improvements:

Study Aim and Hypothesis: Begin with a clear statement of the study aim and the hypothesis that ABO blood group associations with COVID-19 severity might be mediated through specific biological mechanisms.

Results Presentation: Use comparative language to clarify differences between groups. For instance, specify that patients with A/AB blood groups showed higher ICU mortality (29% vs. 11%) and more frequent ventilation needs (71% vs. 52%).

Limitations and Future Research: Mention the retrospective nature and limited sample size as key limitations in the abstract, along with a note on the need for further studies to confirm these associations and investigate underlying mechanisms.

Introduction and Background

The introduction provides useful context on ABO blood groups and general COVID-19 pathology but would benefit from a clearer statement of the study’s specific hypothesis and objectives related to endothelial and adipocyte activation.

Consider adding recent references to further strengthen the literature review, particularly regarding the protective effects of anti-A antibodies in B/O groups and the potential role of thrombotic risk factors associated with blood groups.

Methods Section

Clarify Patient Selection and Data Collection:

Explicitly state inclusion and exclusion criteria and provide more details on data collection timing and handling to enhance reproducibility.

Consider including information on quality control for biomarker assays and any adjustments made for potential confounders (e.g., comorbidities, treatment variations).

Rationale for Group Stratification:

Justify the stratification into A/AB vs. B/O groups by explaining the relevance of anti-A antibodies and thrombotic risk differences.

Consider the limitations of combining blood groups in this way and address how it may affect the interpretation of findings.

Statistical Analysis:

Provide more details on statistical methods used, including reasons for selecting non-parametric tests and any adjustments made for confounding factors.

Consider including sensitivity analyses to confirm the robustness of findings.

Results Section

Standardize Data Presentation:

Use consistent y-axis scales and bar lengths across figures to improve readability.

Present summary statistics (e.g., medians, interquartile ranges, odds ratios) in tables for clarity.

Emphasize Key Findings with Clear Comparative Language:

Use clear language to highlight significant results, such as ICU mortality and ventilation differences, and visually distinguish statistically significant values in tables and figures.

Provide visual aids, like bar charts or box plots, to represent biomarker comparisons across groups, as these will improve readers’ ability to interpret differences.

Discuss Non-Significant Findings and Trends:

Briefly mention trends or non-significant findings, particularly in biomarker analyses, to give a balanced view of the results and suggest areas for future investigation.

Discussion Section

International Comparisons:

Discuss the results in the context of international studies, such as those in other populations or regions, to assess whether these findings may be generalizable. Specific studies include:

https://doi.org/10.2174/1389203723666220811121803

https://doi.org/10.3389/bjbs.2022.10098

10.1097/MD.0000000000028334

https://doi.org/10.1016/j.transci.2021.103169

Highlight differences that may arise from genetic, demographic, or environmental factors.

Interpret Key Findings in Light of Hypotheses:

Relate findings directly to the study’s hypotheses about anti-A antibodies and thrombotic risks associated with blood groups. Discuss any unexpected or null findings in biomarkers to provide a comprehensive view.

Acknowledge Limitations and Implications for Future Research:

Discuss limitations, such as the sample size, observational nature, and potential confounding factors. Suggest that future studies with larger, diverse populations and expanded biomarker analyses could further explore these associations.

Highlight Clinical Implications:

Briefly discuss how these findings could inform COVID-19 risk assessment or ICU resource allocation if validated in larger cohorts.

Ethical Considerations

Dual Publication: No concerns about dual publication were noted, provided that the study’s content is original and not previously published in a similar format elsewhere.

Data Confidentiality: Ensure that patient confidentiality is maintained, with no identifying information in figures or descriptions.

Research Ethics Approval: Confirm that ethical approval was obtained for this study and explicitly mention it in the methods section. If applicable, discuss patient consent, particularly if data were collected retrospectively.

Conclusion Section

Summarize Key Findings and Limitations:

Provide a focused summary of the main findings, with a cautious interpretation due to the study’s limitations.

Avoid overstating results and emphasize that further research is required to confirm these associations.

Suggest Specific Directions for Future Research:

Include specific recommendations for future studies, such as investigating ABO blood group associations with COVID-19 in larger, multi-center studies and exploring specific biomarkers in diverse populations.

References Section

Update with Recent Literature:

Ensure references are updated with the latest literature on ABO blood groups, COVID-19 severity, and relevant biomarkers.

Include Contrasting Studies for a Balanced Perspective:

Where possible, add studies with findings that contrast with or complement this study to present a balanced view of the literature.

Ensure Consistent Formatting:

Confirm that the formatting and accuracy of citations meet journal guidelines.

Reviewer #2: Association of ABO Blood Group with Endothelial and Adipocyte Activation in COVID-19

Introduction:

Role of markers of endothelial injury and adipocyte activation has not been introduced. These are the primary parameters of study that should be highlighted thoroughly.

It looks like a discussion, moreover severity, risk factors, markers should by emphasized.

Methods:

“as previously described.” Should be removed from the line no. 167.

Inclusion and exclusion criteria not clear.

Time period of patient recruited is vague. Some where its written from march 2020, to march 2024 and in tables its upto march 2021.

What was the criteria used for severity?

Results:

Exact p value should be written.

Comparison of the ABO blood groups with severity should be highlighted before their association with the clinical factors, even though author compared in previous studies but this should be included in supplementary.

Steroid, antiviral, immunomodulatory drugs, anti-inflammatory used for no. of days should be highlighted and change in the levels of markers should be highlighted as it is a cohort.

Is there any effect of drugs used on adipocyte modification and endothelial injury?

Except from the markers do author seen any changes in the endothelium or epithelium?

Levels of the sIL-6R, sgp130, others markers should be compared with the severity and clinical parameters especially comorbid and drugs. These are compared using the regression model so p value should be highlighted in the figures.

Discussion:

Justification of findings should be emphasized with proper references.

Reviewer #3: 1. “Some 85 studies, including one from our own center in British Columbia, Canada, have shown 86 that blood group A is associated with more severe disease, and blood group O with less 87 severe disease, [6-8] while others have not found an association [9]” , because you mentioned “others have not found” I think it is better to add more references which mentioned it. I saw some unbalance number between references which mentioned an association and no association.

2. Related to the hypotheses of “the anti-A theory”, I understand that patients with type O and type B have anti-A antibodies, but the previous paragraph only mentioned about type O and I can’t find the explanation about type B such as what kind of diseases are less or more susceptible to this blood type B.

3. The “criteria” for 128 patients, does it mean ARDS criteria in line 188 or ICU criteria or severe COVID-19 criteria?

4. In line 295 about “Serum cytokines and IL-6 signaling receptors”, you mentioned that no differences in the circulating levels of this cytokine or IL-6 signaling receptors, and you mentioned some references IL-6 signaling receptors which might not explained much about the association of ABO. Don’t you think it is better if you compare with healthy patients just like what your references did? Because your study compared between anti-A antibodies group vs no anti-A antibodies.

5. If it is possible I want to get clearer explanation and summary of “Serum cytokines and IL-6 signaling receptors”

6. You mentioned “Adipsin and leptin are elevated in children with COVID-19” but your sample’s average age is more than 50 years old. Can you summarize or explain whether adult or elder people have the same phenomena or not?

7. Since your data were accessed from Mar 30, 2020 to Sep 30, 2021, do you think the SARS-CoV-2 variant influenced the data of patient severity?

8. I understand that the sample size indeed is small compared to the duration of your accessed data, what limits the data collection process so that the sample size is small?

9. In line 234-236, you mentioned about the percentage of blood type of your sample, does this percentage numbers also consistent with the percentage of blood type in normal situation in Canada?

10. From two hypotheses mentioned in line 119-120, in your opinion, which hypotheses is supported by your study data?

Reviewer #4: Although the small sample size, the article reflects good knowledge and important to the field of the blood groups and COVID-19. Here are some comments to be addressed:

Line 97: Although it was written in italics, please make it clear that you are mentioned "the ABO gene".

Line 180: Define BMI.

Line 191: Define PTT.

Line 218: Define IQR.

Line 304: Please remove the repeated word "elevated".

Overall, very nice work!

Reviewer #5: - Did you consider checking Rhesus factor among your population?

- If available, Could you elaborate more on low d-dimer levels in discussion part.

- For steroids, Was steroids started before or after ICU admission? What was the agent and dosing?

- No tocilizumab alone group? or your guidelines mandates that tocilizumab should be started with of after steroids?

- If applicable, it would be bitter to include outcomes on the incidence of thrombosis or bleeding.

6. PLOS authors have the option to publish the peer review history of their article (what does this mean? ). If published, this will include your full peer review and any attached files.

**Do you want your identity to be public for this peer review?** For information about this choice, including consent withdrawal, please see our Privacy Policy .

Reviewer #1: **Yes: ** Saeed M Kabrah

Reviewer #2: **Yes: ** Dr. Santosh Kumar Sidhwani

Reviewer #3: **Yes: ** Anna Lystia Poetranto, DVM., Ph.D

Reviewer #4: **Yes: ** Amr J Halawani

Reviewer #5: No

---

## [Author Response · Author response to Decision Letter 1]

12 Feb 2025

see attached word document - Response to reviewers.

---

## [Editor Report · Decision Letter 1]

16 Feb 2025

ABO Blood Group and COVID-19 Severity: Associations with Endothelial and Adipocyte Activation in Critically Ill Patients

PONE-D-24-45720R1

Dear Dr. Y.C. Chen,

Thank you for submitting the revised version of the manuscript. Having examined it and the accompanying response-to-review document, I find that the concerns and suggestions raised by the five referees of the original manuscript have been satisfactorily addressed in the revised version.

I am therefore pleased to inform you that your manuscript has been judged scientifically suitable for publication and will be formally accepted for publication once it meets all outstanding technical requirements.

Kind regards,

Santosh K. Patnaik, MD, PhD

Academic Editor

PLOS ONE
---

## [Editor Report · Acceptance letter]

PONE-D-24-45720R1

PLOS ONE

Dear Dr. Y.C. Chen,

I'm pleased to inform you that your manuscript has been deemed suitable for publication in PLOS ONE. Congratulations! Your manuscript is now being handed over to our production team.

Kind regards,

on behalf of

Dr. Santosh K. Patnaik

Academic Editor

PLOS ONE